# Wear, Osteolysis, and Aseptic Loosening Following Total Hip Arthroplasty in Young Patients with Highly Cross-Linked Polyethylene: A Review of Studies with a Follow-Up of over 15 Years

**DOI:** 10.3390/jcm12206615

**Published:** 2023-10-19

**Authors:** Christopher F. Deans, Brandt C. Buckner, Kevin L. Garvin

**Affiliations:** Department of Orthopaedic Surgery & Rehabilitation, University of Nebraska Medical Center, 985640 Nebraska Medical Center, Omaha, NE 68198, USA

**Keywords:** highly cross-linked polyethylene, total hip arthroplasty, young arthroplasty, polyethylene wear

## Abstract

Total hip arthroplasty (THA) has in recent years trended toward a younger, more physically demanding patient population. Mid- to long-term studies of all ages of THA patients using highly cross-linked polyethylene (HXLPE) have been favorable, but concerns about its long-term failure and wear-related complications remain for young THA patients. In this narrative review, a search of the PubMed/MEDLINE and Cochrane databases was performed, and we identified six studies with a minimum 15-year follow-up of HXLPE with various femoral head materials. Wear-related revisions were exceedingly low for patients under the age of 55, with variable reports of non-clinically significant osteolysis. Higher activity levels, a larger femoral head size, and femoral head material were not associated with greater long-term wear rates. Young THA with metal or ceramic on HXLPE is exceedingly durable with favorable outcomes at follow-ups of over 15 years.

## 1. Introduction

Total hip arthroplasty (THA) has been called the “operation of the century”, demonstrating itself to be both a clinically successful and cost-effective procedure for the management of hip arthritis [1]. Using US National Inpatient Sample and Census Bureau data, models have predicted a 129% increase in THA procedures by 2030, amounting to approximately 850,000 procedures [2]. With this increase in primary THA procedures, particularly as younger, more active patients seek primary THAs, revision THA procedures in the US have been projected to increase by between 43% and 70% by 2030 [3]. Between 2010 and 2018 in South Korea alone, approximately 25,000 revision THAs were performed, with projections of an increase in the incidence of revision THAs by 1.1% per year until 2030 [4]. In Romania between 2001 and 2022, approximately 11,000 revision THAs were performed, and projections estimate a doubling of revision THA volume by 2034 [5]. This has required arthroplasty surgeons to adjust for a patient population that may demand a higher level of activity and longer implant survivorship compared to an older patient population [6,7].

There have been many reports describing THA’s clinical success at long-term follow-ups [8,9,10,11]. Cementless acetabular fixation has been shown to be more durable than cemented acetabular fixation at 20 years [10,12]. In appropriately selected patients, metaphyseal-fitting cementless femoral stems demonstrate an excellent survival rate at up to 23 years [11,13]. While these developments in improved fixation have been employed to increase long-term survivorship in young adults undergoing THAs, concerns regarding the durability of the bearing surface in terms of wear, the development of osteolysis, and loosening persist.

Historically, long-term implant survivorship in young THA patients was limited by polyethylene wear, in which particulate debris cause osteoclastic-mediated bone resorption, periprosthetic osteolysis, and eventual aseptic loosening. This osteolysis problem had been identified as the leading cause of the late-term failure of total joint arthroplasties [14,15]. Multiple studies describing the ten-year results of THAs with conventional polyethylene (CPE) reported implant survival rates as low as 83% and osteolysis rates up to 56% [16,17,18,19,20,21].

Highly crosslinked polyethylene (HXLPE) was first introduced in the late 1990s in an effort to decrease wear and periprosthetic osteolysis, effectively increasing long-term implant survivorship. This has resulted in promising mid-term wear rates and a decreased burden on revision THA procedures secondary to polyethylene wear [22,23,24,25]. A recent meta-analysis of compiled studies in all age groups with 5- to 15-year follow-ups found a 4.1 to 6.6% osteolysis rate with a 0 to 5.2% risk of hardware failure requiring revision for wear [22]. Multiple studies reporting follow-ups up to 12 years in patients less than 50 years of age report a 0% revision rate for wear or osteolysis [23,24,25].

Many cohort studies and randomized controlled trials have explored various HXLPE liners regarding their wear rates, functional outcomes, radiographic osteolysis, and revision rates [18,19,20,21,22,23,24,25,26,27,28,29,30,31,32]. However, most of these were mid-term follow-up periods in patient cohorts of all ages. The aim of the present review was to analyze reports of long-term (≥15 years) follow-ups of patients with THA who were ≤55 years. Specifically, we sought to compare various reports describing wear rates, the development of radiographic osteolysis, the incidence of implant failure for aseptic loosening, and revision rates secondary to wear-related diagnoses. In addition, we evaluated multiple variables, including the femoral head material, femoral head size, polyethylene processing, and activity level, as well as their effect on wear rates and wear-related complications. 

## 2. Materials and Methods

For this narrative review, searches of the PubMed/MEDLINE and Cochrane databases were performed on 7 July 2023, utilizing the following search terms for English-language publications: (total hip arthroplasty OR THA OR total hip replacement) AND (crosslinked polyethylene OR cross-linked polyethylene OR cross-linked polyethylene OR XLPE) AND (young OR < 55 years OR age < 55). 

Articles underwent title and abstract screening. Screened articles were read in full, and their inclusion was agreed upon by 2 authors (C.F.D. and B.C.B.). All studies with patient cohorts >55 years or with mean follow-up of <15 years were excluded, as well as articles that were not available in the English language. Ceramic-on-polyethylene, metal-on-polyethylene, and oxidized zirconium-on-polyethylene were all included. Articles including second-generation HXLPE with vitamin E infusion were reviewed and commented upon in this manuscript. However, their outcomes were not included with review of first-generation HXLPE due to the limited length of follow-up term. A large amount of registry data were also reviewed and commented upon in this manuscript. However, their outcomes were also not included in combination with other studies due to the heterogeneity of age inclusions. 

## 3. Results

Six studies were identified describing follow-ups over a minimum of 15 years for HXLPE with various femoral head materials in 676 patients (745 THAs) with a mean age at index surgery of less than 55 years [4,33,34,35,36,37].

### 3.1. Polyethylene Wear Analysis

Wear was reported in five of the six studies. Steady-state wear, described as linear wear measured after a 1-year bedding-in period, was reported to be between 0.01 and 0.04 mm/year in five studies [4,33,35,36,37]. Volumetric wear was reported to be between 4.50 and 19.43 mm^3^/year in four studies [33,35,36,37]. These were measured from X-rays using different methods including evaluations via Martell Hip Analysis Suite (Ver 8.0.4.3), PolyWare (Draftware Developers, Warsaw, IN, USA), and AutoCAD (2013) [4,33,35,36,37]. See Table 1. 

Rames et al. in 2021 evaluated 22 patients (26 THAs) with a mean age of 38.5 years and a mean follow-up of 16 years, reporting the lowest linear and volumetric wear rates at 0.01 mm/year (SD 0.05) and 4.50 mm^3^/year (SD 5.80), respectively [35]. Conversely, Roedel et al. in 2021 evaluated 95 patients (105 THAs) with a mean age less than 50 years and a mean follow-up of 17.3 years, finding the highest linear wear at 0.04 mm/year (SD 0.02) [36]. Youngman et al. in 2023 reviewed 115 patients (128 THAs) at a mean age of 38 years, finding the highest volumetric wear at 19.43 mm^3^/year (SD 35.16) [37]. 

### 3.2. Osteolysis, Revision Rates, and Implant Survival

All six studies reported on osteolysis, revision rates, and implant survival. Four of the six studies used x-rays only for the evaluation of osteolysis, while the other two studies reported x-ray and computed topography (CT) findings. See Table 1. The revision rates due to polyethylene wear or osteolysis ranged from 0% to 0.01%, while the overall revision rates ranged from 0 to 8.6% [4,33,34,35,36,37]. A single revision was reported by Youngman et al. for aseptic femoral loosening requiring revision [37]. Three revisions for stem failure were described by Bryan et al. [34]. Two of these were described as aseptic loosening and one as stem breakage. These aseptic loosening instances were not reported as secondary to polyethylene wear due to there being no radiographic signs of osteolysis or polyethylene wear [34]. The remaining revisions were for non-wear or loosening-related diagnoses including: 15 instabilities (2.0%), 12 infections (1.6%), and a leg length discrepancy (0.001%) [4,33,34,35,36,37]. The survivorship with revision for wear or osteolysis as the endpoint was 99.2 to 100% at a minimum of 15 years [4,33,34,35,36,37].

Osteolysis adjacent the acetabular or femoral component via a radiographic analysis was reported at a rate of 0 to 11.5% [4,33,34,35,36,37]. Via a CT scan, the rate increased from 0 to 34.6% [4,35]. Only one of these patients underwent a revision procedure for osteolysis, with the remainder undergoing clinical monitoring with no radiographic or clinical signs of implant loosening. Rames et al. reported 9 of 26 THAs with osteolysis; 9 of these were identified by CT scans only, and 3 of those were able to be seen on radiographs. Those seen on radiographs were primarily in Zone 1 or 2 of the acetabulum-adjacent posterosuperiorly-placed acetabular screw [35,38]. Sixty-six percent of identified instances of osteolysis were located in the retroacetabular space and required CT scans for their identification but were clinically not significant [35]. Rames et al. reported that patient cohorts separated by the presence of osteolysis and absence of osteolysis reported no significant difference in patient reported outcomes (PROs), implant survival, or cup inclination and anteversion [35]. Multiple studies noted limited radiolucent lines around the acetabulum, without any radiographic sign of progression or implant loosening [35,36,37].

A recent study based on data from the Australian registry found the cumulative revision rate of HXLPE to be 6.2% for all ages, and 6.6% for the age group ≤ 55 years with a mean follow-up of 16 years [39]. For comparison, the revision rate for the patients ≤ 55 years with CPE was 17.4%. The Swedish registry in 2013 showed that the 12-year follow-up with HXLPE revision rate was 1.9% for any reason compared to 4.3% for CPE [40]. A recent study using data from the National Joint Registry of the United Kingdom showed that the rate of HXPLE at 13 years was 1.5% compared to 3.6% for CPE [41]. Additionally, this study showed the highest rate of revision in patients under the age of 55, but did not stratify this group based on polyethylene type. 

The New Zealand Orthopaedic Association Joint Registry’s 24-year report determined that the annual revision rate per 100 implants was 0.49 for ceramic with XPLE compared to 0.81 for CP. Similarly, the rate per 100 implants for XPLE on metal was 0.50 compared to 0.79 with CP. They also reported higher rates of revision for patients under 55, but did not compare polyethylene type within age groups.

### 3.3. Risk Factors for Wear

All six studies reported on the various risk factors for wear previously described using conventional polyethylene (CPE).

Femoral head material was not shown to significantly impact wear rates in these studies. Three studies’ cohorts all included cobalt chromium (CoCr), two studies’ cohorts all comprised ceramic (alumina, BIOLOX-forte, CeramTec, Plochingen, Germany), and a single study included both [4,33,34,35,36,37]. Roedel et al. perform ed acomparative analysis of the wear rates of CoCr, oxidized zirconium (Oxinium, Smith & Nephew), and ceramic, finding no significant difference between ceramic (0.03 mm/year linear; 7.35 mm^3^/year volumetric), CoCr (0.04 mm/year; 6.50 mm^3^/year), and oxidized zirconium (0.04 mm/year; 5.53 mm^3^/year) [36]. 

Femoral head size has not been shown to affect wear rates in the included studies. The femoral head sizes ranged from 22 mm to 32 mm, with the vast majority being 28 mm and largely ceramic or CoCr. Roedel et al. performed a direct comparative analysis of wear rates based on femoral head size, finding no significant differences between 32 mm (0.03 mm/year linear; 6.44 mm^3^/year volumetric), 28 mm (0.04 mm/year; 6.45 mm^3^/year), and 26 mm heads (0.03 mm/year; 6.44 mm^3^/year) [36].

Activity level based on patient reported outcome surveys was described in four of the included studies. A significant improvement in Harris hip scores (HHS) was reported in three studies and the University of California, Los Angeles Activity Score (UCLA) in three studies. Rames et al. evaluated the difference in wear rates between cohorts of very active patients (UCLE 8-10) and less active patients (UCLA 1-7), finding no significant difference in linear wear (0.030 mm/year vs. 0.016 mm/year; *p* = 0.310) or volumetric wear (16.01 mm^3^/year vs. 12.38 mm^3^/year; *p* = 0.543) [33]. Kim et al. and Youngman et al. also reported the UCLA scores, with pre-operative means of 2 and 4 followed by post-operative means of 7 and 6, respectively [4,37]. While these post-operative UCLA scores would each be considered in a less-active cohort based on Rames’ criteria, their cohorts’ wear rates remain low despite an increase in their baseline activity levels.

HXLPE processing has been a topic of discussion regarding differing wear rates. A recent study based on the National Joint Registry of the United Kingdom revealed polytheylene liners irradiated with at least 50 kGy had a better survival rate at 14 years, but that highly irradiated polyethylene (defined as 100 kGy or more) provided no additional benefit in the revision and wear rates of relatively moderately irradiated polyethylene (defined as 50 to 100 kGy) [41]. Four studies included the use of the Zimmer Longevity HXLPE (Zimmer-Biomet, Warsaw, IN, USA), two studies used the Depuy Marathon HXLPE (DePuy, Leeds, UK), and a single study included both [4,33,35,36,37]. The Zimmer Longevity HXLPE was clinically introduced in 1998, manufactured with GUR 1050 sheet compression polyethylene resin, radiated via an electron beam with 100 kGy at approximately 40 degrees Celsius. It is then remelted at 150 degrees Celsius for 6 h, followed by gas plasma sterilization. The Depuy Marathon HXLPE was also introduced in 1998, manufactured with a GUR 1050 extruded rod, radiated with 50 kGy. This is then remelted at 155 degrees Celsius for 24 h, followed by re-annealing at 120 degrees Celsius for 24 h, and sterilized in gas plasma. Similar wear rates are noted amongst all six studies with two different liners, each of which have been irradiated with at least 50 kGy. Unfortunately, Bryan et al. did not perform a direct comparative analysis of the wear between the Longevity and Marathon liners [34]. It is notable that the CT evaluation by Rames et al. demonstrating rates of osteolysis up to 34.6% using the Longevity liner is in contradiction to the 0% rate of CT-identified osteolysis found by Kim et al. using the Marathon liner [4,35]. The presence of osteolysis did not have clinical significance.

## 4. Discussion

Our purpose was to review the current evidence on the longest-term outcomes of HXLPE in young (<55 years) THA patients. Specifically, we reviewed the wear rates, development of radiographic osteolysis, and revision risk for wear-related diagnosis, as well as multiple other variables’ effects on the wear. We found that although there are conflicting reports on the development of CT-determined osteolysis, this seems to be clinically insignificant. The long-term wear rates and survival of THA in young patients using HXLPE remain favorable at up to 21 years. 

Survival-free all-cause revisions using HXLPE are superior to the long-term outcomes of metal-on-metal (MoM) and ceramic-on-ceramic (CoC) bearings. While MoM bearings have demonstrated hardly any wear and generate a very small volume of wear debris, the presence of adverse local tissue reactions (ALTRs) and metallosis is well known and has resulted in the decreased utilization of MoM constructs. Reports with follow-ups up to 28 years of McKee-Farrar THAs have shown an implant survivorship of 74% [42,43]. The fourth-generation CoC survival for any-cause revisions has been reported to be 95.9% to 96.7% at follow-ups up to 18 years [44,45]. Alshammari et al. reviewed 235 patients with follow-ups up to 18 years, reported five revisions including one for squeaking, two infections, one stem loosening, and two stem fractures for a 96.7% survivorship rate [45]. Similarly, Blummenfeld reviewed 186 THAs with follow-ups up to 10.5 years, also reporting five revisions, including three for liner fractures, one recurrent dislocation, and one pain and squeaking for a cumulative survivorship rate of 95.9% [44]. The included studies in this report have shown an overall survivorship rate of 91.4 to 100% at up to 21 years, with the lowest survivorship reported by Youngman et al. secondary to instability and infection [4,33,34,35,36,37].

National registry reports have also shown good overall revision rates for HXPLE on ceramic or metal compared to CPE, MoM, and CoC articulations. The New Zealand Orthopaedic Association Joint Registry’s 24-year report reports an annual revision rate/100 components of 0.48 for CoC and 1.38 for MoM, compared to 0.49 for ceramic on HXPLE and 0.50 for metal on HXPLE. Accordingly, the Swedish registry and the National Joint Registry in the UK both report similar lower revision rates for XPLE (1.9% at an average follow-up of 12 years and 1.5% at an average follow up of 13 years, respectively) when compared to CPE (4.3% and 3.6%, respectively) [40,41]. The New Zealand Orthopaedic Association Joint registry also reported an improved survival rate of XPLE with annual revision rates/100 implants that are over 1.5 times lower than those for CPE. These three registries report higher revision rates for patients under the age of 55 compared to older patients but do not specify polyethylene type. The Australian registry reported a cumulative revision rate of HXLPE of 6.2% for all ages and 6.6% for the age group ≤ 55 years with a mean follow-up of 16 years [39]. In that registry, the revision rate for CPE was over 2.5 times higher at 17.4% for patients under the age of 55. Given the similar revision rate in the XPLE ≤ 55 group compared to all ages, the higher revision rates in the younger populations in the other registry studies could be due to CPE. Further stratifications of the data for patients aged ≤55 between XPLE and CPE may further clarify this in future registry studies. 

Various factors have been evaluated and are associated with increased wear in conventional polyethylene; however, these associations have not been supported in HXLPE mid-term studies or this report’s included long-term studies. Rames and coworkers astutely evaluated activity level as a risk factor for wear in young THA patients, finding no significant difference between very active patients (with UCLA scores of 8–10) and less active patients (with UCLA scores < 8) at 15 years [33]. While these activity-related sub-analyses were not explicitly performed by other studies in this review, the results of Kim and Youngman demonstrated a significant increase in patient activity pre-operatively to post-operatively with no apparent effect on wear [4,37]. In a shorter-term report, Guy et al. reported on 34 ceramic-on-HXLPE THAs in patients with UCLA scores of 9–10 and involved in impact activities, including activities such as skiing, heavy labor, and running. They reported a linear wear rate of 0.03 mm/year (+/−0.01 mm/year) at the minimum 5-year follow-up with no radiographic signs of osteolysis or loosening [46]. This is well below the wear rate supported to be a risk for the development of osteolysis and is consistent with the wear rates in this review [36,47,48].

Femoral head material and size have been theorized to affect polyethylene wear rates, in part due to in vitro studies demonstrating greater friction with metal and ceramic on polyethylene than MoM or CoC [49,50]. Roedel and coworkers directly compared wear rates up to 21 years in their report between metal, ceramic, and oxidized zirconium femoral heads at various sizes. An independent statistical analysis demonstrated no significant difference in the HXLPE wear rates with different femoral head materials or femoral head sizes up to 32 mm [36]. This is consistent with other mid- to long-term outcomes in patient cohorts of all ages [51,52]. Lachiewicz et al. compared linear and volumetric wear rates in different femoral head diameters on HXLPE at a mean of 11 years. Their cohort of 84 hips with CoCr heads with a diameter from 26 to 40 mm did not demonstrate any difference in wear rates based on femoral head size [52]. More recently, Thalody and coworkers evaluated the minimum 10-year outcomes of large (32 or 36 mm) ceramic or CoCr femoral heads in small (48 or 50 mm) acetabular cups. They reported low linear and volumetric wear rates (0.04 mm/year, 39.60 mm^3^/year) with no significant differences in wear rates between head sizes or materials [51]. Furthermore, the Australian registry has demonstrated cumulative revision rates over 10 years to be similar for oxidized zirconium, ceramic, and CoCr femoral heads on HXLPE. It seems that the femoral head bearing material and size may be chosen based on alternative factors, such as hip stability, implant cost, and availability, although ceramic does offer a theoretical advantage over CoCr due to its greater scratch resistance, in turn reducing the risk of abrasive wear, and its reduced risk of corrosion at the head–neck taper [53,54].

These studies have also demonstrated wear rates below what has been widely accepted for the risk of developing radiographically evident osteolysis. Historical studies have shown thresholds of linear wear rates > 0.10 mm/year and a volumetric wear of >150 mm^3^/year associated with an increased risk of developing osteolysis; however, these are based on conventional ultra-high-molecular-weight polyethylene (UHMWPE) liners and evaluated via radiographs rather than advanced imaging, such as CT [47,48]. Concerns still exist regarding the long-term results of HXLPE due the risk of late osteolysis even in the setting of lower wear rates, theoretically secondary to the different bioreactivity profile of HXLPE wear particles [55,56]. This theoretically increased bioreactivity profile of HXLPE has been refuted in more recent in vitro studies looking at wear particle size and functional biologic activity under physiologically relevant concentrations of protein in joint simulator lubricant [57,58]. The potentially increased bioreactivity of HXLPE particles causing osteolysis has previously not been demonstrated in mid-term HXLPE CT-based studies [32,59]. More recently, Rames et al. evaluated 26 THAs under the age of 50 at a mean follow-up of 16 years via CT for radiographic evidence of osteolysis, finding 34.6% (9 of the 26 THAs) of the patients had osteolysis [35]. This is one of only two discovered reports looking at CT-based evaluations of osteolysis in the longest-term HXLPE THAs, with the other being in direct contradiction to this. Kim et al. in 2020 reported on a 53-patient cohort with a mean follow-up of 17.1 years, finding zero episodes of CT-discovered osteolysis, more consistent with other CT-based midterm studies [4,32,59]. Despite the high rate of osteolysis, Rames et al. noted that this did not correspond with wear-related problems or revisions [35].

The current practice of maximizing femoral head size comes at the expense of using a thinner liner. Concerns have been raised about HXLPE oxidation and the risk of late fatigue failure due to the sacrifice of mechanical properties for greater cross-linking [60]. With no reports of liner failure at up to 21 years, this review suggests that the maximization of the head-to-liner ratio is acceptable for young THAs. As the hip is a highly conforming joint, sacrifices of mechanical properties, such as the tensile and yield strength as well as the fatigue crack propagation resistance, are acceptable in pursuit of improved wear resistance [61]. However, reports of mechanical failures of HXLPE exist. Wahl et al. reports on an early failure of HXLPE for an adverse reaction to polyethylene particles, found to be secondary to inadequate irradiation during manufacturing [62]. Waewsawangwong and coworkers reported on a failure at 20 months using CoCr on HXLPE due to a crack at the rim of the liner’s locking mechanism. They deduced that the 65-degree abduction angle and 2.3 mm liner thickness at the site of failure were strong contributors and recommended a minimum polyethylene thickness of 6 mm [63]. Roesler et al. reported on a liner failure 13 years after an index THA and 2 years after a multilevel lumbar spinal fusion. This patient’s liner failed at the rim, which was suspected to be due to the lack of pelvic motion during positional changes and repetitive neck–liner impingements [64]. Ast et al. reviewed 70 cases of liner failure from a single manufacturer derived from voluntary reporting to the United States Food and Drug Administration (FDA). Their findings supported the importance of cup positioning, ensuring no internal impingement, and recommended a minimum rim polyethylene thickness of 4.7 mm [65]. While the included long-term studies of this report do not demonstrate any instances of fatigue failure, these case reports and FDA reviews are compelling. It is imperative that the recommendations for appropriate implant positioning which avoids internal impingements, for a weight-bearing minimum liner thickness of 6 mm, and for a rim minimum liner thickness of 4.7 mm be adhered to in order to minimize the risk of mechanical failure. 

Among the limitations of this review was the small number of studies satisfying our search criteria with mild heterogeneity in the aforementioned studies. For example, the measurement of wear was performed using several different methods and computer-based measurement platforms. Each of these come with their own standard deviation in the measurement error. The reader must keep this in mind when interpreting these results. We have sought to remedy this limitation by reporting the measurement methods of each study, as can be seen in Table 1. Additionally, we are at the will of each studies’ authors regarding the thoroughness of their reporting of wear-related imaging findings, complications, and revisions. We omitted multiple patient-related and surgery-related factors that may impact HXLPE wear, such the patient body mass index (BMI) and cup positioning. As these were not consistently reported in the accepted studies, we elected not to include them. 

While the use of HXLPE in young THA patients provides an unrivaled degree of wear resistance, there are several directions of focus for future studies. 

Long-term studies mostly include patients with smaller femoral heads than is now the trend. Longer-term outcomes using >36 mm diameter heads and thin liners will help us to better understand and maximize the stability when using a larger femoral head, while minimizing the risk of mechanical failure using thin liners.

Concerns about the late oxidation of HXLPE have led to further innovations with the addition of antioxidants, such as vitamin E or alpha-tocopherol [60,66]. Early in vivo results have been favorable with retrieval studies describing inhibited oxidative processes and reduced free radicals, as well as clinical studies demonstrating promising early-term survival and wear rates [67,68,69]. Longer-term outcomes will help us understand if the clinical outcomes are superior to those of antioxidants added to HXLPE.

CT-detected osteolysis in HXLPE has now been described in up to 34% of cases [35]. Clinical decision-making algorithms for follow-up intervals, imaging modalities, and indications to revise an asymptomatic hip with osteolysis prior to catastrophic failure are limited due to a lack of knowledge of which lesions will progress and the risk factors of progressions. The continued reporting of long-term outcomes of HXLPE patients with asymptomatic osteolysis will help us provide further guidance.

While most hip surgeons continue to place at least one acetabular screw in their implanted cups, Rames et al. demonstrated via CT that, although clinically insignificant, most osteolysis abutted screw tracts [35]. The innovation of highly porous cup coatings may now allow surgeons to minimize the effective joint space while maximizing the newly implanted cup’s stability by avoiding screws in the majority of patients.

## 5. Conclusions

The longest-term outcomes of HXLPE THAs in young patients demonstrate low wear rates and favorable implant survivorship. However, the continued monitoring of this patient cohort for late wear-related complications is necessary to provide further guidance as THAs continue to be performed in younger patients.

## Figures and Tables

**Table 1 jcm-12-06615-t001:** Summary of studies with ≥ 15-year follow-up in patients aged ≤55 years.

Study (Year)	No. of Patients (No. of Hips)	Bearing Type	Mean Age at Index Surgery (Years)	Length of Follow-Up (Years)	Number Revised (%)	Wear Measurement Method	Bedding-in Period, Years	Steady State Wear, mm/year (SD)	Volumetric Wear, mm^3^/year (SD)	Complications
Rames et al. (2019) [33]	82 (89)	CoCr on Zimmer Longevity HXLPE ^1^	38.8	15	2.2%	- Martell Hip Analysis Suite, Ver 8.0.4.3 - XR	1	0.019 (0.05)	12.80 (22.69)	2 infection w/revision 0 lysis
Bryan et al. (2019) [34]	237 (273)	CoCr on Zimmer Longevity ^1^ or Depuy Marathon ^2^	42.3	16	5.1%	NR	NR	NR	NR	6 instability w/revision 5 infection w/revision 3 stem failure w/revision 0 lysis
Kim et al. (2020) [4]	133 (133)	Ceramic on Depuy Marathon HXLPE ^2^	53	17.1	3%	- Auto CAD 2013 - XR - CT	NR	0.016 (0.03)	NR	2 infection w/revision 2 instability w/revision 0 lysis
Rames et al. (2021) [35]	22 (26)	CoCr on Zimmer Longevity HXLPE ^1^	38.5	16	0%	- Martell Hip Analysis Suite, Ver 8.0.4.3 - XR - CT	1	0.01 (0.05)	4.50 (5.80)	9 lysis w/o revision
Roedel et al. (2021) [36]	87 (96)	CoCr, ceramic, Oxinium on Zimmer Longevity HXLPE ^1^	41.6	15 to 21	1	- PolyWare, Ver 8 - XR	NR	0.04 (0.02)	6.22 (7.06)	1 instability w/revision 0 lysis
Youngman et al. (2023) [37]	115 (128)	Ceramic on Zimmer Longevity HXLPE ^1^	38	16	8.6%	- Martell Hip Analysis Suite, 8.0.4.3 - XR	NR	0.019 (0.07)	19.43 (35.16)	6 instability w/revision 3 infection w/revision 1 implant failure w/revision 1 leg length discrepancy w/revision

SD, standard deviation; NR, not reported; CoCr, Cobalt Chrome. ^1^ Zimmer Longevity HXLPE. 1998 clinical introduction. GUR 1050 sheet compression resin. 100 kGy e-beam at ~40 deg C radiation. Remelted at 150 deg C for 6 h. Sterilized in gas plasma. ^2^ DePuy Marathon HXLPE. 1998 clinical introduction. GUR 1050 ram extrusion resin. A total of 50 kGy γ-ray at RT radiation. Remelted at 155 deg Celsius for 24 h, followed by re-annealing at 120 deg C for 24 h in reduced O2. Sterilized in gas plasma.

## Data Availability

No new data has been created in the production of this manuscript.

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
