# Peer review of "Wear, Osteolysis, and Aseptic Loosening Following Total Hip Arthroplasty in Young Patients with Highly Cross-Linked Polyethylene: A Review of Studies with a Follow-Up of over 15 Years"

_jcm, 2023, doi:10.3390/jcm12206615_

Round 1

Reviewer 1 Report

The authors reviewed over 15-year follow-up studies of THAs with HXLPE liners focusing on wear and aseptic loosening. The topic is highly relevant as THA has finally reached a status in which wear is no longer a primary concern, thanks to the introduction of HXLPE from the late 1990s onward. Careful scientific studies, like the present one, are welcome to bring this message to the widest possible readership. A publication is recommended. The present reviewer has only a few comments, which can be considered to require Minor Revision.

Please add title to Table 1.

Discussion, page 6. It is misleading to present a wear rate as 0.03059 ± 0.0084 mm/y. With respect to the actual measurement accuracy, this has too many digits. A preferred way is 0.03 ± 0.01 mm/y. Please check the numeric values throughout the manuscript for consistency.

Discussion, page 6. The superiority of ceramic over metal heads has nothing to do with friction. Friction studies have not shown consistently lower friction values for ceramic. The superiority is based on the higher hardness of ceramic which makes it highly scratch resistant. Abrasion to metallic heads occurs more easily, which leads to increased polyethylene liner wear.

Discussion, page 7. There are mixed results about the biological reactivity of HXLPE wear particles compared with that of CPE particles. Please state this and cite available studies accordingly.

Author Response

“Please add a title to Table 1”.

Thank you, as requested a title has been added to revised manuscript.

“Discussion, page 6… Please check numeric values throughout the manuscript.”

We have reviewed numeric values throughout manuscript and, for the sake of consistency, have rounded all numbers to hundredths. While this runs the risk of over-estimating some of the wear figures from reported literature, these numbers remain below described significant threshold to expect clinically significant wear. Therefore, this should not take away from the accuracy of the report.

“Discussion, page 6. The superiority of ceramic over metal heads has nothing to do with friction … “

Point well taken. We have revised our wording in an effort to make this paragraph clearer. While some in-vitro studies have demonstrated greater friction in metal- or ceramic-on-poly, as compared to metal-on-metal or ceramic-on-ceramic, your point regarding ceramic superior scratch resistance compared to metal is an important distinction for in vivo THA that we have communicated to the reader. Please see lines 248.

“Discussion, page 7. There are mixed results about the biological reactivity of HXLPE compared with that of CPE…”

We have addended this paragraph of discussion to better describe the current status of literature with conflicting reports about biological activity. Please see lines 257 - 259.

Reviewer 2 Report

The research aim is to provide a narrative review which analyzes long-term follow-ups of young patients with THA taking into consideration multiple factors.

The abstract is written and structured appropriately.

The introduction transposes the research into the topic and formulates the objective of the study at the end. However, in terms of epidemiology the amount of annual hip revision rates performed in recent years should  be added in relation to other scientific papers, for e.g. Moldovan F, Moldovan L, Bataga T. A Comprehensive Research on the Prevalence and Evolution Trend of Orthopedic Surgeries in Romania. Healthcare (Basel). 2023 Jun 27;11(13):1866. doi: 10.3390/healthcare11131866.

In the methodology section, the search strategy and inclusion of studies are presented. The subsection in this case has no point and should be deleted. The results are clearly described.

The discussions interpret the research results and relate them to other scientific papers. The limitations of this review should be better emphasized at the end of this section.

The conclusions are concise and clear.

The references are adequate but can be extended as suggested above.

Author Response

Reviewer #2

“The introduction … the amount of annual hip revision rates performed in recent years should be added in relation to other scientific papers…”

Thank you for the recommendation. We agree that including annual revision rates of recent years, particularly from different regions of the world for an international audience, strengthens the introduction. We have revised this in reflection of your comments. Please see lines 38 – 41.

“In the methodology section, the search strategy and inclusion… The subsection in this case has no point and should be deleted…”

We appreciate the keen eye. We agree and have deleted this subsection.

“The discussions interpret the research results and relate them… the limitations of this review should be better emphasized at the end of this section.

Thank you, it is certainly important to make note of limitations to the reader. We have amended this, please see lines 289 - 298.

“The references… can be extended as suggested above.”

We greatly appreciate the suggestions, and the references have been updated accordingly.
